

# Integrating chromatin conformation information in a self-supervised learning model improves metagenome binning

Harrison Ho[1,2], Mansi Chovatia[1], Rob Egan[1], Guifen He[1], Yuko Yoshinaga[1], Ivan Liachko[3], Ronan O'Malley[1,4] and Zhong Wang[1,2,4]

[1] Department of Energy Joint Genome Institute, Lawrence Berkeley National Lab, Berkeley, CA, United States
[2] School of Natural Sciences, University of California, Merced, CA, United States
[3] Phase Genomics Inc, Seattle, WA, United States
[4] Environmental Genomics and Systems Biology Division, Lawrence Berkeley National Lab, Berkeley, CA, United States

Corresponding author
Zhong Wang, zhongwang@lbl.gov

## ABSTRACT

Metagenome binning is a key step, downstream of metagenome assembly, to group scaffolds by their genome of origin. Although accurate binning has been achieved on datasets containing multiple samples from the same community, the completeness of binning is often low in datasets with a small number of samples due to a lack of robust species co-abundance information. In this study, we exploited the chromatin conformation information obtained from Hi-C sequencing and developed a new reference-independent algorithm, Metagenome Binning with Abundance and Tetra-nucleotide frequencies—Long Range (metaBAT-LR), to improve the binning completeness of these datasets. This self-supervised algorithm builds a model from a set of high-quality genome bins to predict scaffold pairs that are likely to be derived from the same genome. Then, it applies these predictions to merge incomplete genome bins, as well as recruit unbinned scaffolds. We validated metaBAT-LR's ability to bin-merge and recruit scaffolds on both synthetic and real-world metagenome datasets of varying complexity. Benchmarking against similar software tools suggests that metaBAT-LR uncovers unique bins that were missed by all other methods. MetaBAT-LR is open-source and is available at https://bitbucket.org/project-metabat/metabat-lr.

## INTRODUCTION

The rapid proliferation of high-throughput sequencing in metagenomics, combined with the advancement of scalable computational tools, has allowed scientists to digitally isolate tens of thousands of microbial genomes from large collections of metagenomic datasets (*Nayfach et al., 2021*). Although these genomes are only the tip of the iceberg of microbial diversity, they are instrumental in gaining insight into the genetic and metabolic capabilities of their hosts and understanding their interactions. Several metagenomics

pipelines have been developed to transform short unassembled reads into metagenome-assembled genomes (MAGs), and most of them consist of four stages: read quality improvement, assembly, binning, and quality assessment. Briefly, in the read quality improvement step, redundant information (sequencing adapters, PCR duplication, contaminants) is removed, and sequencing errors are corrected. The reads are then assembled into contigs and scaffolds in the assembly step. In the binning step, contigs/scaffolds that are likely from the same species are clustered into genome bins. Finally, the bins are further polished, verified for completeness and purity, and filtered based on predefined criteria such as the Minimum Information about a Metagenome-Assembled Genome (MIMAG, *Bowers et al., 2017*). For details of this process, please refer to recent reviews (*e.g.*, *Breitwieser, Lu & Salzberg, 2019*; *Ayling, Clark & Leggett, 2020*).

Although both sequencing technologies and software tools are rapidly improving, predicting MAGs from datasets of complex microbial communities with thousands of species remains a challenging process. Binning, for example, could produce low-quality bins, especially with datasets with high strain heterogeneity and high circular DNA content according to a recent comprehensive benchmarking study (*Meyer et al., 2021*). Many automatic metagenome binning software, including metaBAT 2, rely on tetranucleotide frequency (TNF) and abundance covariance between multiple samples (*Kang et al., 2019*) to achieve high accuracy. However, the vast majority of metagenomic datasets have only a few samples, which severely hampered their performance. As a result, the completeness of the MAGs derived from these datasets is quite low, on average between 20% and 40% (*Meyer et al., 2021*). Various strategies have been implemented to solve this problem, from the use of long-read sequencing technology (*Bickhart et al., 2022*; *Frank et al., 2016*), synthetic long reads (*Chen, 2019*), with the use of recent breakthroughs in single-cell metagenomics (*Arikawa et al., 2021*; *Bowers et al., 2022*). Although these strategies were effective in low- or medium-complexity datasets, scaling them to high-complexity communities such as those found in water or soil samples to capture nondominant species is technically challenging and expensive. Recently, the long-range chromatin-level interaction information generated by proximity ligation technologies, such as the Hi-C method, has shown promising results not only in binning but also in linking circular DNAs to their hosts (*Burton et al., 2014*; *Beitel et al., 2014*; *Press et al., 2017*). Compared to the methods mentioned above, Hi-C-based methods have the potential to resolve similar species or even strains, since they rely on experimental evidence derived from physical interactions of genomic regions within the same cell. Furthermore, because millions of cells can be analyzed in a single test tube at a low cost, Hi-C holds the promise of deconvolving genomes from complex microbial communities.

The Hi-C data are inherently noisy (*Yaffe & Tanay, 2011*). Existing binning tools use various strategies to combat these noises. For example, bin3C uses the Knight-Ruiz algorithm to normalize Hi-C read counts between different scaffolds before building a contact map using these reads. Then it uses the Infomap clustering algorithm to group scaffolds into bins (*DeMaere & Darling, 2019*). Phase Genomics ProxiMeta is proprietary software that applies an algorithm based on Markov Chain Monte Carlo on Hi-C linkage information to form metagenome bins (*Press et al., 2017*), and its implementation has not

been described in full detail. The recently published BinSPreader is able to use long-range information such as long reads, or short reads derived from Hi-C experiments, combined with the assembly graph to refine existing binning results (*Tolstoganov et al., 2022*).

The binning problem can be conceptually broken down into a two-step problem that first predicts the likelihood that two scaffolds come from the same genome, followed by a clustering step to form bins. Machine learning methods have previously been developed for the first prediction step, including semi-supervised approaches such as SolidBin (*Wang et al., 2019*) and SemiBin (*Pan et al., 2022*) that take advantage of known reference genomes, or unsupervised approaches such as VAMB based on variational autoencoder (*Nissen et al., 2021*).

Here, we proposed a new strategy to integrate Hi-C information into a self-supervised learning framework to improve the completeness of metagenome binning. The resulting software, Metagenome Binning with Abundance and Tetra-nucleotide frequencies—Long Range (metaBAT-LR), first builds a random forest model from a subset of the initial binning result using features that include the number of Hi-C reads that bridge the two scaffolds, the abundance, and the lengths of the scaffolds. It then predicts the connectivity between all pairs of scaffolds in the dataset and uses the prediction to either merge bins or recruit unbinned scaffolds. Applying it to several datasets with varying complexity, we show that metaBAT-LR can leverage Hi-C reads to improve binning completeness and discover novel bins missed by alternative tools.

## MATERIALS AND METHODS

### Datasets

#### Generating synthetic mock metagenome community Hi-C datasets

The ZymoBiomics Microbial Community Standard (Zymo Research Corp., Irvine, CA, USA), referred to as the Zymo Mock dataset, contains 10 species: *Pseudomonas aeruginosa*, *Escherichia coli*, *Salmonella eterica*, *Lactobacillus fermentum*, *Enterococcus faecalis*, *Staphylococcus aureus*, *Listeria monocytogenes*, *Bacillus subtilis*, *Saccharomyces cerevisiae*, and *Cryptococcus neoformans*. Except *Saccharomyces cerevisiae* and *Cryptococcus neoformans*, which have an abundance of 2%, the abundances of all species are at 12% of total genomic DNA. The genome sizes of the reference genomes are as follows: *Pseudomonas aeruginosa* at 6,738,157 bp, *Escherichia coli* at 17,764,482 bp, *Salmonella eterica* at 18,992,842 bp, *Lactobacillus fermentum* at 8,829,291 bp, *Enterococcus faecalis* at 12,660,551 bp, *Staphylococcus aureus* at 11,846,679 bp, *Listeria monocytogenes* at 16,894,761 bp, *Bacillus subtilis* at 15,028,083 bp, *Saccharomyces cerevisiae* at 12,616,115 bp, and *Cryptococcus neoformans* at 18,284,049 bp.

Eight sequencing libraries (one whole genome shotgun, seven Hi-C) were generated from this synthetic community. Whole genome shotgun (WGS) libraries were generated using standard Illumina sequencing protocols. Two Hi-C libraries, CZWON and GPANX, were made using the Arima-HiC+ kits (Arima Genomics Inc, San Diego, CA, USA), following the manufacturer's protocol. Five Hi-C libraries, CZWOH, GPANY, GPANZ, GPAOA and GPAOB, were created by Proximo Hi-C Microbe kits (Phase Genomics Inc,

Seattle, WA, USA) following the protocol provided by the manufacturer. This data has the NCBI BioProject number of: PRJNA846282. Detailed library creation parameters are shown in the Supplemental Section (Fig. S1). Intermediate data can be found at https://doi.org/10.5281/zenodo.8226572.

### Cat fecal microbiome dataset

The Cat Fecal Microbiome Dataset was a publicly available dataset obtained through ProxiMeta (Phase Genomics, Seattle, WA, USA). The NCBI Accession numbers are the following: SRR22078166 for Hi-C reads and SRR23092418 for the WGS reads. Intermediate data can be found at https://doi.org/10.5281/zenodo.8226572.

### Human fecal microbiome dataset

The Human Fecal Microbiome Dataset is a published data set sequenced using the ProxiMeta Hi-C kit (*Press et al., 2017*). The NCBI BioProject accessing ids are: PRJNA413092, Accession: SRR6131122, SRR6131123, and SRR6131124. This library contained two different Hi-C libraries that were constructed due to the use of two different restriction enzymes (SRR6131122, SRR6131124), MluCI and Sau3AI (New England Biolabs, Ipswich, MA, USA). Intermediate data can be found at https://doi.org/10.5281/zenodo.8226572.

## Preprocessing WGS and Hi-C reads

### Quality and adapter filtering

The sequence adapters from the WGS and Hi-C read libraries were filtered using bbduk from the BBTools software suite (v38.86) (*Bushnell, 2014*) under the following parameters 'k=23 ktrim=r mink=12 hdist=1 minlength=50 tpe tbo'. The reads were then quality-trimmed using the following parameters 'qtrim=rl trimq=10 minlength=50 chastityfilter=True'.

### Metagenome assembly and initial binning

MetaSpades (v3.14.1) was used with default parameters to assemble the WGS reads (*Nurk et al., 2017*). The binning was carried out according to the standard metaBAT 2 protocol: BBTools software suite bbmap (v38.86) was used to map WGS reads to scaffolds (*Bushnell, 2014*). Samtools sort and index were applied to the read alignment data mapped (*Li et al., 2009*). metaBAT 2, with the minimum scaffold length set to 1,500, was then used to create the initial set of metagenome bins.

## The MetaBAT-LR pipeline

The pipeline contains three main steps: Hi-C read alignment, self-supervised learning, and bin merging/scaffold recruitment (Fig. 1).

### Alignment of Hi-C reads to metagenome scaffolds

Each Hi-C read pair was assigned separately to the scaffolds using BWA-MEM with the '-5SP' option (*Li, 2013*). The '-5' was used to reduce the number of secondary and alternate mappings. The '-S' and '-P' options were used to avoid assumptions about mate-pair libraries. Samblaster was used with default settings to remove PCR duplicates (*Faust &*

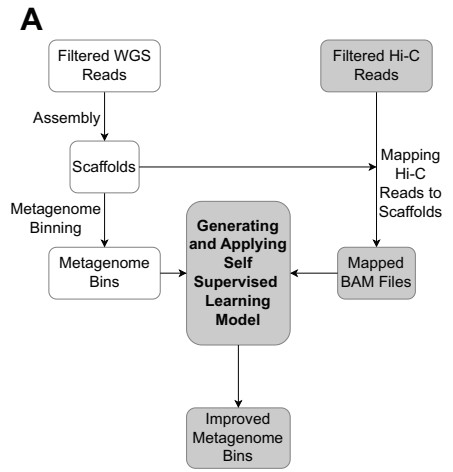

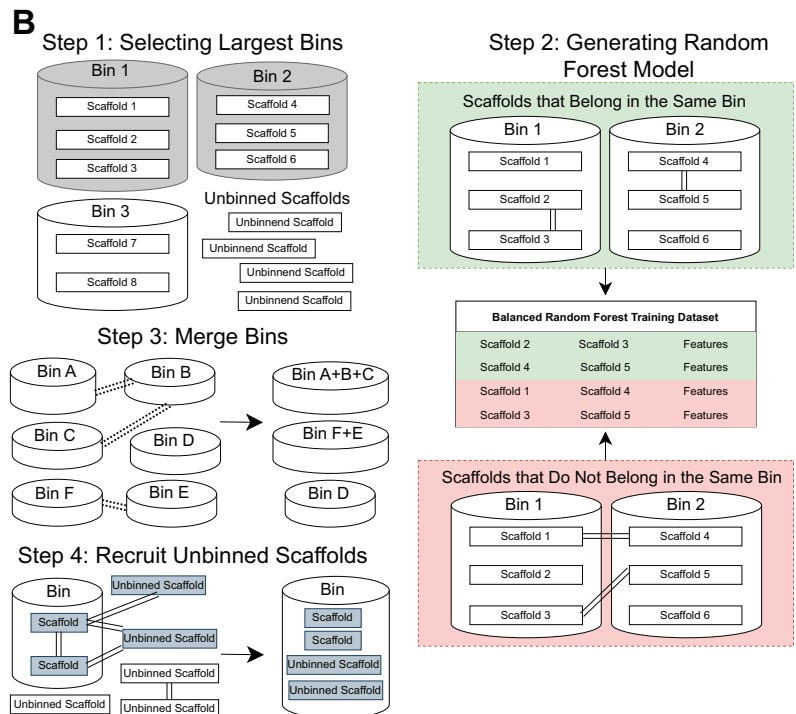

**Figure 1 An overview of the MetaBAT-LR pipeline.** (A) A standard WGS metagenome binning workflow is shown as boxes in white. Filtered WGS reads are first assembled into scaffolds, before being binned into metagenome bins (For details, see 'Methods'). The shaded boxes represent how Hi-C information is integrated to improve binning. First, quality-filtered Hi-C reads are mapped to the scaffolds. A random forest model is subsequently built using features derived from the largest bins (hence self-supervised). Lastly, the model predicts the connectivity between all pairs of scaffolds in the dataset and uses the prediction to either merge bins or recruit unbinned scaffolds to obtain a final set of refined metagenome bins. (B) There are four key steps in generating and applying the self-supervised machine learning model. First, the algorithm will select a specified number of the largest bins in the initial dataset. Then, a random forest model will be trained using data that has been extracted from the largest bins based on the Hi-C read information linking scaffolds from the same bin *vs* scaffolds from different bins. After that, the random forest model will be used to determine if existing bins should be merged together based on the Hi-C read connectivity between the scaffolds that comprise the bins. Lastly, the model will be used, along with LPA, to evaluate Hi-C read connections between scaffolds already placed in bins and unbinned scaffolds to determine if some unbinned scaffolds should be recruited into existing bins.

**A**

| Feature Name | Formula |
|---|---|
| Normalized Hi-C Read Counts | $10^6 \times R/\sqrt{l_1 \times l_2}$ |
| Difference in Sequencing Coverage | $|\log_2(d_1/d_2)|$ |
| Total Sequencing Coverage | $\log_{10}(d_1 \times d_2)$ |
| Variance of Total Sequencing Coverage | $\log_{10}(v_1 \times v_2)$ |
| Total Connectivity | $\log_{10}(c_1 \times c_2)$ |
| Hi-C Reads Connecting Two Scaffolds | R |

**B**

| Normalized Hi-C Read Counts | Difference in Sequencing Coverage | Total Sequencing Coverage | Variance of Total Sequencing Coverage | Total Connectivity | Hi-C Reads Connecting Two Scaffolds |
|---|---|---|---|---|---|
| 17.81 | 0.03 | 2.36 | 2.57 | 2.59 | 2 |
| 8.76 | 4.96 | 3.86 | 4.48 | 4.38 | 4 |
| 19.37 | 0.09 | 2.34 | 2.22 | 2.55 | 2 |

**Figure 2 Features used for the Random Forest Model.** (A) The features used in our random forest model are calculated based on the number of Hi-C reads connecting two scaffolds (R), their lengths ($l_1, l_2$), sequencing coverage ($d_1, d_2$), sequencing coverage variations ($v_1, v_2$), and the total number of connections each scaffold has ($c_1, c_2$). (B) Example of the features used to train the random forest model.

*Hall, 2014*). Finally, Samtools View was used with the options '-S -h -b -F 2316' to filter out reads that are not mapped, not the primary alignment, or are secondary alignments.

### Self-supervised machine learning

A subset of the initial bins generated by metaBAT2 is selected to generate a training dataset. As mentioned above, the predominant problem with these bins is incompleteness and not contamination; larger bins are more likely to have higher completeness. Therefore, we select the largest bins (10 by default) to derive a data matrix for training. Each pair of scaffolds within the same genome bin is labeled "1", while scaffold pairs from different bins are labeled "0"s. As the pairs labeled "0"s may be inaccurate due to the incompleteness of the bins, they are assigned a smaller weight than those labeled "1"s. For each pair of scaffolds, several features are calculated based on the counts of Hi-C reads connecting them (R), their lengths ($l_1, l_2$), sequencing coverage ($d_1, d_2$), sequencing coverage variations ($v_1, v_2$) and the number of connections each scaffold has ($c_1, c_2$). Specific features are detailed in Fig. 2. The feature normalized Hi-C read counts includes the constant $10^6$ as a multiplication factor as a way to avoid losing precision.

Random forest is used (RandomForestClassifier from the Scikit-Learn library) to build a machine learning model to classify whether or not a pair of scaffolds are from the same genome.

### Bin merging and scaffold recruitment

The machine learning model built previously is applied to all possible pairs among all scaffolds to obtain the probability that two scaffolds are in the same genome, regardless of whether they are in the training set or not. The percentage of pairs connected between two bins, the depth score vector, and the cosine similarity of their tetranucleotide frequency (TNF) vectors are then used to select pairs of bins that should be merged. The formal mathematical notation of these metrics is shown below. Each bin can be represented by a node. Pairs of connected bins are used as edges to build a graph of bins to get partitions, with all bins within a partition predicted to be derived from the same genome.

Given a set of bins $B = \{b_1, b_2, \ldots, b_k\}$, with each $b_i$ containing a set number of scaffolds, represented by $b_i = \{s_{i1}, s_{i2}, \ldots, s_{it}\}$. We use $num(b_i)$ to represent the number of scaffolds in bin $b_i$. We define $pair(s_{im}, s_{jn})$ as a function representing whether there is a connection between the scaffold $s_{im}$ and $s_{jn}$, where $s_{im} \in b_i$, $s_{jn} \in b_j$, and $i \neq j$. More specifically, $pair(s_{im}, s_{jn}) = 1$ if $prob(s_{im}, s_{jn}) \geq minscore$, otherwise $pair(s_{im}, s_{jn}) = 0$. The probability $prob(s_{im}, s_{jn})$ is determined by the random forest model, and $minscore$ is defined by the user or defaulted to 0.5. The LR_score is defined as:

$$LR\_score(b_i, b_j) = \frac{\sum_{m=1}^{num(b_i)} \sum_{n=1}^{num(b_j)} pair(s_{im}, s_{jn})}{num(b_i) \times num(b_j)} \tag{1}$$

Let $depth(s_{im})$ denote the average depth of scaffold $s_{im}$, where $s_{im} \in b_i$. The vector of the average scaffold depth for bin $b_i$ can be expressed as: $D(b_i) = \{depth(s_{i1}), depth(s_{i2}), \ldots, depth(s_{it})\}$.

Let $D(b_i)$ and $D(b_j)$ be the vectors of average scaffold depths for bins $b_i$ and $b_j$, respectively. Then, the depth score is defined as the $p$-value of the t-test for bin pair $b_i$ and $b_j$ and can be expressed as:

$$depth\_score(b_i, b_j) = p\_value(D(b_i), D(b_j)) \tag{2}$$

Let $T(b_i)$ and $T(b_j)$ be the vectors of TNF for bins $b_i$ and $b_j$, respectively. The cosine similarity score can be expressed as:

$$tnf\_score(b_i, b_j) = cosine\_similarity(T(b_i), T(b_j)) = \frac{T(b_i) \cdot T(b_j)}{||T(b_i)|| \; ||T(b_j)||} \tag{3}$$

where $\cdot$ denotes the dot product, and $|| \; ||$ denotes the norm.

To merge incomplete bins, we built a graph with bins as nodes, edges were determined by a pair of bins that have LR_score, depth_score, and tnf_score above certain thresholds. Then the connected component algorithm was used to merge the bins.

To recruit unbinned scaffolds, a graph is built with all scaffolds, both binned and unbinned, represented as nodes and their predicted probability of being the same genome as edges. This predicted probability is determined by the random forest model. We offer two algorithms to partition the graph, The label propagation algorithm (LPA) or Louvain, with LPA used as the default (*Raghavan, Albert & Kumara, 2007*). Unbinned scaffolds are recruited to a bin if the partition contains only one known bin.

## Applying MetaBAT-LR to the synthetic and realworld datasets

The initial assembly/partitioning was carried out using 20% of the WGS dataset. An independent set of 10% WGS reads was used as a negative control for the Hi-C datasets. The Hi-C datasets were also sampled to match the number of reads in the negative control. The completeness of the metagenome bins was calculated by mapping the bins to reference genomes using MetaQuast (*Mikheenko, Saveliev & Gurevich, 2016*).

To study the Hi-C read depth needed for metagenome binning, we chose the CZWOH library (105,367,232 reads) and subsampled it at a percentage gradient ranging from 10% to 90% in increments of 10%, and also at 5%. Each subsampling experiment was replicated five times.

Both the Cat Fecal and Human Fecal datasets were assembled and binned using the methods listed above. A contamination filter of 10% was applied to the bins before and after the MetaBAT-LR pipeline.

## Hi-C metagenome binning performance comparison

BinSPreader v3.16.0.dev (*Tolstoganov et al., 2022*), bin3C v0.1.1 (*DeMaere & Darling, 2019*), hicSPAdes (*Ivanova et al., 2022*), and ProxiMeta Web version (*Press et al., 2017*) were run with default settings. The final metagenome bins generated by these software tools were compared with those of metaBAT-LR. As BinSPreader is a metagenome bin refiner, the same initial metagenome bins used as inputs for metaBAT-LR, were also used as initial input bins for BinSPReader. For the Zymo Mock Synthetic Dataset, AMBER v2.0.3 was used to evaluate and compare the performance of the five tools (*Meyer et al., 2018*). For the Cat Fecal and Human Fecal datasets, dRep v2.0.0 (*Olm et al., 2017*) was used to compare the performance of the five methods. dRep was used to determine whether two bins are the same. Using dRep, only bins that had an average nucleotide identity (ANI) score of 97% or higher were considered the same. Furthermore, only medium-quality bins or above were used for comparison (contamination <10% and completeness rate ≥50% (*Bowers et al., 2017*)). A list of all commands-line options of tools used can be found in Fig. S2.

# RESULTS

## Hi-C data increases binning completeness of a single sample dataset from a synthetic mock community

We tested the performance of metaBAT-LR on a dataset derived from a synthetic mock community (Zymo Mock, see 'Materials and Methods'). As the genome sequences of the 10 species within this community are known, this dataset would provide an accurate measure of the quality (completeness and contamination) of every member genome. Additionally, we could afford to sequence this simple community to greater depth, allowing exploration of the relationship between Hi-C sequencing depth and binning performance.

From a set of 15 initial bins obtained 20% of the WGS reads from the GPANZ synthetic dataset, we selected the three largest bins to train a random forest model. From the information in these three bins, we created a balanced dataset with 91 pairs of scaffolds.

We then randomly selected 70%, 15%, and 15% of these pairs as training, validation, and testing datasets to build random forest models, respectively. The resulting model has a test accuracy of 0.992 (precision: 0.994, recall: 0.997) (GPANZ 5,761,142). The high precision rate indicates that the model accurately discriminates scaffolds from different genomes. The lower recall rate indicates that it misses some connections between scaffolds of the same genome. This is expected because not all parts of a genome physically interact with each other and not all interactions could be captured by Hi-C. The impact on binning is likely low because a pair of scaffolds can be indirectly linked by a third scaffold of the same genome. We observed similar results in all Hi-C datasets in addition to the GPANZ dataset, suggesting that the six selected features mentioned above are sufficient to predict scaffold-scaffold connections.

Although we used only 20% WGS to form the initial bins and created a "sensitized" scenario to observe the effect of Hi-C, simply adding 10% more WGS reads had little effect on genome completeness (Fig. 3). In contrast, except for 4/10 species that already have completeness greater than 90%, adding the same amount of Hi-C reads led to a dramatic increase in percent genome completeness for four of the remaining five species (an increase in percent genome completeness from 26% to 72% for *Lactobacillus fermentum*, 45% to 95% for *Bacillus subtilis*, 51% to 91% for *Enterococcus faecalis*, and 37% to 74% for *Saccharomyces cerevisiae*, respectively). The Eukaryotic genome that showed little improvement, such as *Cryptococcus neoformans*, have much larger genome sizes and lower abundance than others (at 2% *vs* 12% for the other eight bacterial species), indicating poorer assemblies due to low sequencing depth. When metaBAT-LR is applied to bins that do not contain contamination, metaBAT-LR is able to increase genome completeness without increasing contamination (Fig. 3B left set of bar plots). In the cases where the original metagenome bins do have some contamination, metaBAT-LR generally does not increase the contamination. This is exemplified by *Escherichia coli*. *Saccharomyces cerevisiae* is a rare case in which contamination from the original metagenome bin caused the contamination rate to increase after the Hi-C information was integrated. By reviewing the total aligned length information generated by Quast, we are able to dissect why this occurred (Fig. S3). Most *Saccharomyces cerevisiae* scaffolds were clustered in bin 15. Bin 15 contains a small amount of contamination from scaffolds from *Cryptococcus neoformans*. The two organisms are different types of yeast and have low DNA material, at 2%, from the synthetic community sample. When metaBAT-LR is applied, many unbinned *Cryptococcus neoformans* scaffolds are correctly recruited to bin 15 because there are no other bins that contain *Cryptococcus neoformans* scaffolds. This, in turn, increases the level of contamination for bin 15. This is why the contamination of *Saccharomyces cerevisiae* is so high in the figure. Further analysis from AMBER calculated the F1 score (bp) to be 0.951, while the average purity was at 0.955.

In addition, both bin merging and scaffold recruitment observed in the genomes described above are dependent on the Hi-C sequencing depth. The analysis did not show significant differences in genome completeness depending on the number of Hi-C reads used, with a median genome completeness of around 95% for all cases. More Hi-C reads did not appear to confer more benefits (Fig. S4).

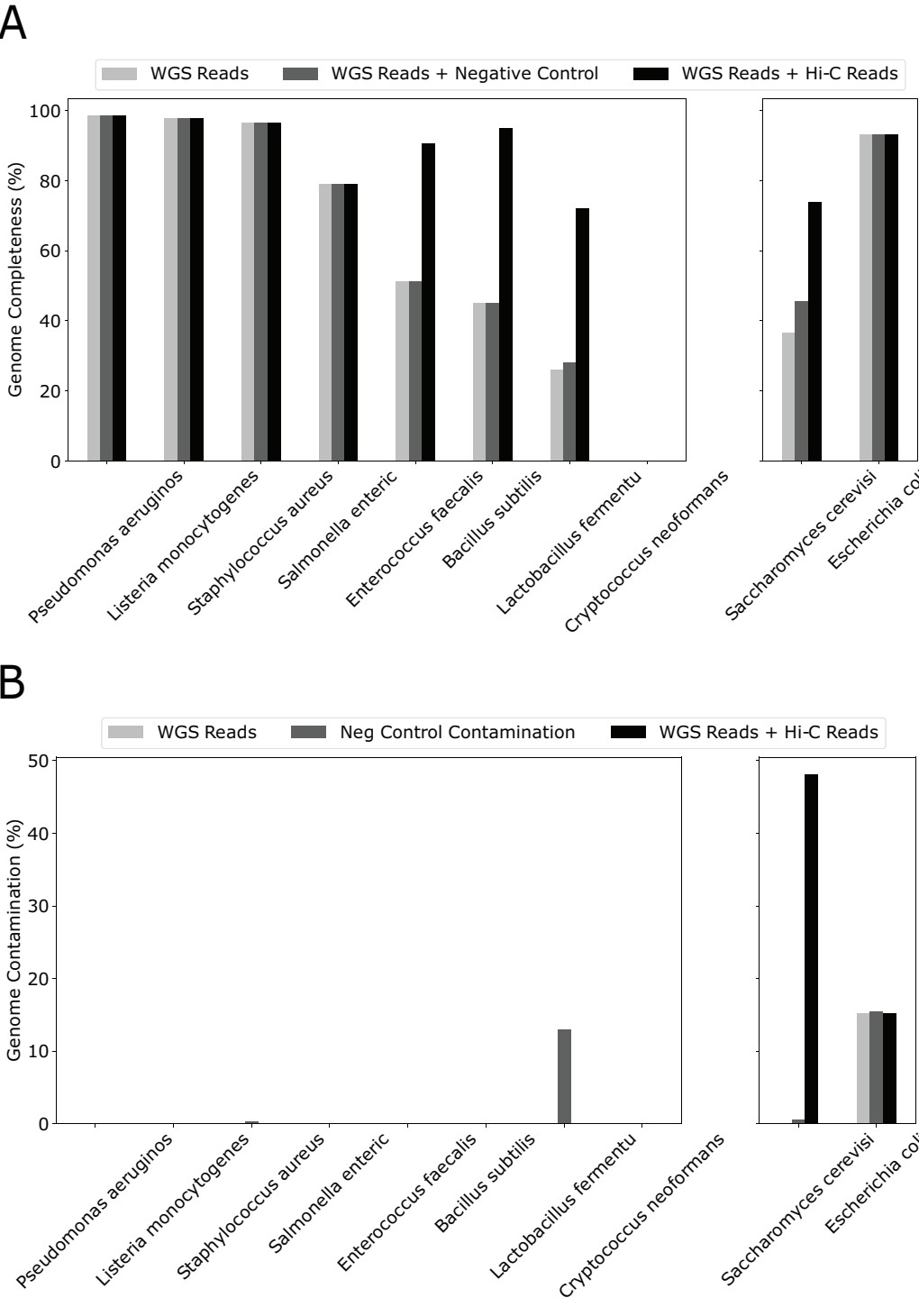

**Figure 3** **Hi-C reads improve genome completeness over the initial metagenome bins on a synthetic community.** (A) The effect on genome completeness for the 10 genomes in the Zymo Mock community by WGS Reads (light gray bars), WGS Reads + Negative Control (medium gray bars), or WGS Reads + Hi-C Reads (dark gray bars). The 10 genomes are on the x-axis, and genome completeness is on the y-axis. The two genomes on the right contained traces of contamination before metaBAT-LR was applied. (B) The change in genome contamination for the 10 genomes in the Zymo Mock community.

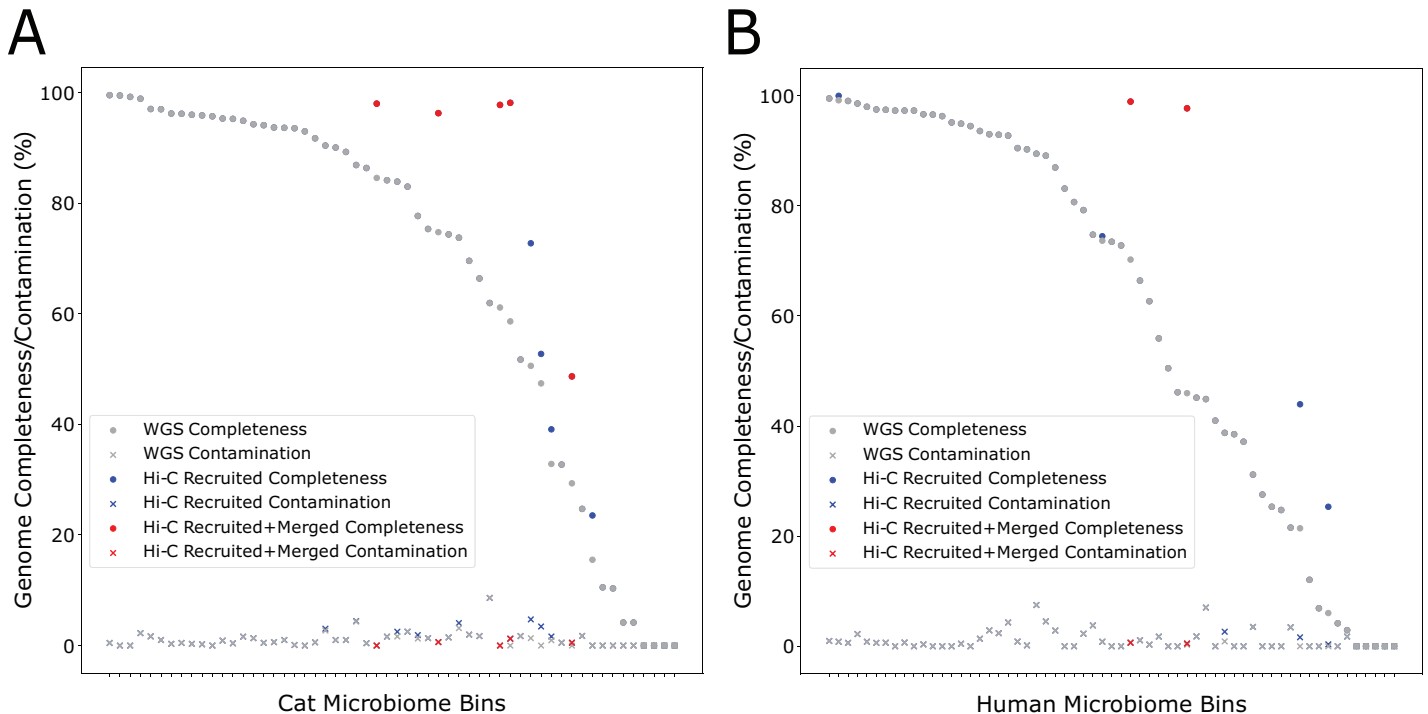

**Figure 4** **Hi-C data improves binning results for the Cat (A) and Human (B) Fecal microbiome datasets.** The estimated genome completeness is shown on the y-axis, and bins are sorted according to their completeness before metaBAT-LR was applied in descending order on the x-axis. Each dot represents a bin. A dot in gray shows the completeness value of a particular bin before metaBAT-LR, and the blue dot with the same x-coordinate shows the completeness after Hi-C-mediated scaffold recruitment. In some cases, a red dot with the same x-coordinate is also shown to designate the completeness after both Hi-C-mediated scaffold recruitment and bin merging.

We performed similar experiments by varying different Hi-C sequencing kits and different laboratory protocols (Fig. S1). The results of all data sets show an improvement in genome completeness to various degrees (Fig. S5).

## Hi-C data improves binning completeness on real-world datasets

Next, we tested metaBAT-LR on two real-world metagenome datasets with different complexity to further validate its ability to recruit scaffolds and merge bins. As there are no reference genomes available for these datasets, checkM was used to estimate binning performance ('Materials and Methods').

During the self-supervised learning part of the program, 13 high-quality metagenome bins were used to create a balanced machine learning dataset of 5,308 scaffold pairs from the Cat Fecal dataset. The resulting model has a higher accuracy than the synthetic data set (0.961, with a precision of 0.943 and a recall of 0.980). Among the initial 60 metagenome bins, four were filtered after having a high contamination rate from the initial Metabat 2 binning step. Of the 56 bins generated by metabat-LR and filtered for contamination, five new bins were formed when metaBAT-LR merged 2 sets of metagenome bins, increasing the completeness of the bins by an average of 36% (Fig. 4A). In both sets of metagenome bins that were merged, the merging was of two original metagenome bins merging into 1. Scaffold recruitment increased the completeness of four metagenome bins, while the rest of

the bins remained the same. In general, the median genome completeness increased by 4% when Hi-C reads are integrated. At the same time, the median contamination did not increase significantly as a result of scaffold recruitment or bin merging and remained below 1%. This statement still holds even if we had included the four metagenome bins that were filtered out at the beginning of the analysis.

We observed similar results from the human fecal dataset. Among the initial set of 72 metagenomic bins, 11 bins were removed by the contamination filter. Of the remaining bins, two bins were formed when four sets of bins were merged in pairs, resulting in an increase in completeness of an average of 27% among them (Fig. 4B). These six sets of bin merges were the result of merging two distinct bins. The genome completeness of four bins increased due to scaffold recruitment, while the rest showed no change. The median genome completeness of all bins increased by 6%, while the median contamination rate remains below 1%. When examining the overall contribution of scaffold recruiting and bin merging to the increase in genome completeness, we found that bin merging tends to have a much larger effect.

## Fine tuning the number of bins needed to train the random forest model

While the default number is to use the 10 largest metagenome bins to train the random forest, it is a variable that users can change based on their own dataset. If too few bins are used for the training, the model that is generated might not be as accurate. If too many bins are used in the training, there is a chance that the model will be overfit. Furthermore, requiring more bins to be used in training might force metaBAT-LR to train on bins that are not high quality.

We tested a range of between five and 25 bins to be used for training in the human fecal dataset, as it was the most complex, before using CheckM to assess the completeness of the genome and the contamination of the bins. As the number of bins used for training increased, there was a minimal increase in genome completeness in a small number of metagenome bins, but this also increased the contamination rate of those same bins. There were a total of 12 metagenome bins affected by the number of bins used for training. The increases in completeness and contamination can be found in Fig. S6.

## MetaBAT-LR has comparable binning performance to alternative methods with a unique set of bins

MetaBAT-LR adopts a different algorithm from two other popular software tools based on Hi-C, bin3C (*DeMaere & Darling, 2019*) and ProxiMeta (*Press et al., 2017*), as well as the newly published BinSPreader (*Tolstoganov et al., 2022*) and hicSPAdes (*Ivanova et al., 2022*). We compared the binning performance of these five software tools on the Zymo Mock Synthetic dataset, the CAT Fecal microbiome dataset, and the Human Fecal microbiome data sets. One important thing to note is that bin3c and BinSPreader do not merge existing bins, which may influence genome completeness. We used AMBER to evaluate the four methods that were applied to the Zymo Mock Synthetic dataset (Figs. 5A and 5B). When looking at metagenome bins with less than 10% contamination and various
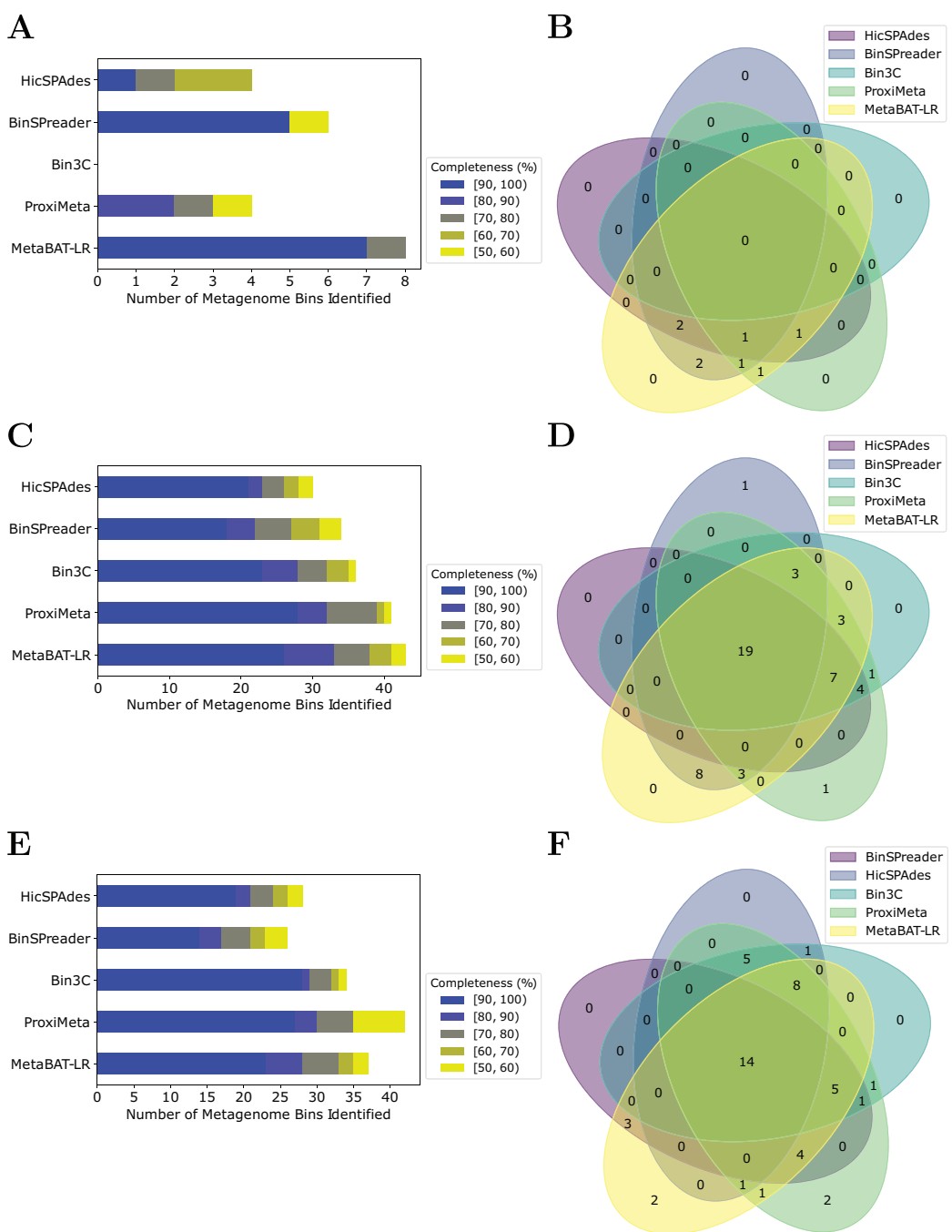

**Figure 5 A comparison of four different Hi-C binning methods.** Information for (A) and (B), was generated using AMBER. Only medium-quality bins (bins with a contamination rate of less than 10% and genome completeness of 50% or greater) were used for comparison. (C–F) Were generated using information from dRep. This was done with a 97% ANI threshold for two bins to be considered the same genome. We also used dRep to impose a 200,000 bp minimum length. We also compared only bins that are of medium quality. The stacked bar graphs (A, C, E) give details about the number of bins found at varying genome completeness levels for each of the methods when applied to the Zymo Mock Synthetic, Cat Fecal, and Human Fecal datasets respectively. The three Venn diagrams (B, D, F) show the number of overlapping bins found for the Zymo Mock Synthetic, Cat Fecal, and Human Fecal datasets respectively.

levels of completeness, MetaBAT-LR was able to find the highest number of metagenome bins at above 90% genome completeness and the highest number of bins overall. A detailed AMBER (*Meyer et al., 2018*) evaluation of these bins can be found in Fig. S7. AMBER-generated graphs summarizing the evaluations can be found in Fig. S8. The Quality of Bins graph shows that MetaBAT-LR results are the closest to the gold standard. Furthermore, the F1 score and average completeness score (bp) of metaBAT-LR, at 0.951 and 0.947 respectively, which was the highest among all tools. The method that came closest to matching metaBAT-LR was binSPreader, which had an average completeness score of 0.674 and an F1 score of 0.800. The other methods were much lower.

In the case of the Cat Fecal and Human Fecal microbiome datasets, we used dRep to match the corresponding bins across the three outputs and CheckM to filter out low-quality bins ('Materials and Methods'). The final count of bins of medium quality or greater (≥50% completeness, <10% contamination) was used as a metric to compare the performance of three tools. In both real-world datasets, there is a significant portion of the bins that are shared among the five tools. In the Cat dataset (Fig. 5B), 19 are shared between all five, while the five together formed 50 bins (Fig. 5D). No tools found all 50 bins, while BinSPreader and ProxiMeta each found a single unique bin that was not found by the others. MetaBAT-LR was able to find the highest number of bins, 43 bins. This was followed by ProxiMeta, which produced 41 bins, bin3C at 37, BinSPreader at 34, and hicSPAdes at 30. A breakdown of the completeness level of these bins can be found in Fig. 5C. In the Human dataset, 14 bins are shared among all five methods, while the five together formed 48 bins (Fig. 5F). MetaBAT-LR and ProxiMeta each found two unique bins that were not found by the others. The unique bins found by metaBAT-LR can be found in Fig. S9. MetaBAT-LR was able to find the 37 bins in total. This is slightly less than the 42 bins found by ProxiMeta but higher than the 34 bins found by bin3C, the 28 bins found by hicSPAdes, and the 26 bins found by BinSPreader. The completeness levels of these bins can be found in Fig. 5E. Tables detailing the exact number of bins found at each completeness level for all three datasets can be found in Fig. S10.

## DISCUSSION

New types of sequencing data are being generated to overcome the low completeness of metagenome bins generated from short-read assemblies with few samples, including Hi-C, long-read sequencing, and single-cell sequencing (*DeMaere & Darling, 2019; Bickhart et al., 2022; Arikawa et al., 2021; Bowers et al., 2022*). Although these new data have shown great promise in improving binning quality, they often increase the analytic complexity. Furthermore, it is challenging to integrate these heterogeneous datasets into a single framework. In this study, we presented a framework based on self-supervised machine learning to take advantage of such datasets. As a proof of concept, we show that it can effectively leverage Hi-C information to improve genome completeness in both synthetic and real-world datasets of various complexity. It should be straightforward to add other types of data as features into the random forest model in metaBAT-LR, or multiple replicates of the same type of data.

The primary goal of this research is to demonstrate that long-range data, such as Hi-C, can be utilized within a robust machine learning framework to enhance metagenome binning outcomes. This is also why we chose not to benchmark our method against numerous metagenome binners that do not incorporate Hi-C reads. If the assembly/binning were perfect, *i.e.*, using the gold standard, then additional Hi-C information will not be able to improve genome completeness among the bins. We have observed varying performances of different binners on different datasets (as seen in the CAMI2 challenge; *Meyer et al., 2021*). We anticipate that metaBAT-LR has the potential to improve the binning experiments of all metagenome binners since it is not dependent on the output of any specific binning tool (we demonstrated this with MetaBAT 2 as an example).

One limitation of our method is metaBAT-LR's dependence on having good Hi-C read-pair connectivity. Sometimes called informative read pairs, these are Hi-C read pairs that are able to map to different contigs. Our method depends on these types of read pair in order to make judgments about recruiting contigs or joining bins. We use the tool hic_qc (https://github.com/phasegenomics/hic_qc) to help determine whether the data set in question has enough Hi-C read connectivity for metaBAT-LR to work.

Although metaBAT-LR could use Hi-C data to improve the completeness of all four incomplete bacterial genomes ($< 75\%$) in synthetic datasets, many incomplete genome bins in the two real-world data sets did not receive any improvement in completeness. There could be several reasons. It is likely that CheckM incorrectly reports genome completeness for unknown species, or Hi-C experiments may not work well for these species. It is also likely that machine learning models have higher false negative rates on real-world datasets, since bin3C and/or ProxiMeta have better results for some bins. An ensemble approach leveraging all five tools can partially alleviate this limitation. One alternative reason for the failure to improve completeness for more bins within the real-world datasets, might be the impact of excluding portions of the metagenomic assembly during the metaBAT 2 binning process. The default settings for metaBAT 2 are to ignore scaffolds under the length of 1,500 bp. The omission of these scaffolds could make it harder to improve individual bins of more complex communities.

Overall, this is a generic method that works not only on Hi-c reads but also on any other types of sequencing read that contains long-range information. The way in which this algorithm takes advantage of the adaptability of random forest models is what allows the overall algorithm to be flexible in application. Further research can be done to test whether other long-range sequencing technologies are compatible. Furthermore, the potential of using long-range information to reduce contamination in the original binning experiment is also worth exploring.

## ACKNOWLEDGEMENTS

The authors thank Dr. Satria Kautsar for his critical review of the manuscript.

### Funding

The work conducted by the U.S. Department of Energy Joint Genome Institute, a DOE Office of Science User Facility, is supported by the Office of Science of the U.S. Department of Energy operated under Contract No. DE-AC02-05CH11231. Ivan Liachko was supported by NIH grants 1R44AI172703 and 5R44AI162570. There was no additional external funding received for this study. The funders had no role in study design, data collection and analysis, decision to publish, or preparation of the manuscript.

### Grant Disclosures

The following grant information was disclosed by the authors:
U.S. Department of Energy Joint Genome Institute.
DOE Office of Science User Facility.
Office of Science of the U.S. Department of Energy operated: DE-AC02-05CH11231.
NIH: 1R44AI172703 and 5R44AI162570.

### Competing Interests

Ivan Liachko is an employee and director of Phase Genomics, Inc., a company that develops Hi-C-based tools. All other authors declare no conflict of interest.

### Author Contributions

- Harrison Ho analyzed the data, prepared figures and/or tables, authored or reviewed drafts of the article, and approved the final draft.
- Mansi Chovatia analyzed the data, authored or reviewed drafts of the article, and approved the final draft.
- Rob Egan analyzed the data, authored or reviewed drafts of the article, and approved the final draft.
- Guifen He performed the experiments, authored or reviewed drafts of the article, and approved the final draft.
- Yuko Yoshinaga performed the experiments, authored or reviewed drafts of the article, and approved the final draft.
- Ivan Liachko conceived and designed the experiments, authored or reviewed drafts of the article, and approved the final draft.
- Ronan O'Malley conceived and designed the experiments, authored or reviewed drafts of the article, and approved the final draft.
- Zhong Wang conceived and designed the experiments, authored or reviewed drafts of the article, and approved the final draft.

### Data Availability

The short reads data is available at NCBI BioProject PRJNA846282.
The code and testing dataset are available at Bitbucket and Zenodo:
- https://bitbucket.org/project-metabat/metabat-lr.

- Ho, Harrison, & Wang, Zhong. (2023). metabat-lr v1.0.0 (1.0.0). Zenodo. https://doi.org/10.5281/zenodo.8040682.

## Supplemental Information

Supplemental information for this article can be found online at http://dx.doi.org/10.7717/peerj.16129#supplemental-information.

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
