# Peer review of "Integrating chromatin conformation information in a self-supervised learning model improves metagenome binning"

_PeerJ, doi:10.7717/peerj.16129_

## Round 0.1 · original submission · Major Revisions

1. For the reader with non-computational background, the authors should consider including a supplementary method with line-by-line commands and exact parameters used for the analysis. It will make their work more transparent.
2. Line 186-187; move to the method section.
3. Figure 4; this data should be in a table.
4. The authors should consider citing the appropriate references in the discussion section.

Reviewer 1 ·

Basic reporting

Authors are presenting a technique of metagenomics binning by incorporating the Hi-C information. Using Hi-C information is quite abundant in metagenomics assembly but not predominantly used in metagenomics binning, especially on Long reads.

However, long reads are not directly handled by the tool. Hence, the tool belongs to the domain of contigs/scaffolds binning. The tool name could be misleading with other tools such as Mega-LR/MetaBCC-LR which are long reads binners.

Presentation of the paper is concise. However, literature study is poor. Although some work has been cited, they have not been used in the evaluation of the performance.

Experimental design

In the self supervised machine learning section, authors discuss the features used for the training. Please present the features in a tabular format for clear presentation. Please justify the choice of feature scaling used.

For example, (1.0e+06 abs(log2(d1/d2))) does not make sense without a proper explanation. This is an assymmetric feature, order of d1 and d2 will affect the feature value. Also constant added seem to mask the log ratio, please explain the mathematics behind this (lines 142-148).

I am not fully convinced by the use of term "self-supervised". As this looks like a semi-supervised approach. Authors use labels from MetaBAT2 to train another model. Self-supervised approach should be able to train on its own using feature discrimination. If the method used Hi-C connectivity between scaffolds with assembly graph connections to train their model, that would push it towards a self supervised method. In this case, no labelled information was used. I suggest authors read more on this - https://en.wikipedia.org/wiki/Self-supervised_learning. A solid bioinformatics related example is training ML models where data points are trained using single copy marker genes (they should be in different bins) and assembly graph connections (connected contigs/scaffolds are more likely to be in the same bin). A good read on this would be https://doi.org/10.1609/aaai.v36i4.20388 (published in AAAI 2022).


Authors then combine the predicted "connectedness like" metric from the training with TNF vectors to decide which bins needs to be merged. Does this mean, the whole pipeline merge bins to improve completeness? Please have a look at the Das Tool program (https://www.nature.com/articles/s41564-018-0171-1). This could be a better baseline for benchmarking due to its popularity.

Please explain how the random forest predictions are combined with TNF. Also, please mention the probability threshold used to create the affinity graph.

Lines 159-160 is not clear. LPA algorithm results in a NxN probability matrix. This tells us the probability of a node (n) having a label (l). How did you use this to partition the graph?

Validity of the findings

lines 186-192 - higher test accuracy could potentiall mean an overfitting scenario. It might be more useful for the reader to evaluate the accuracy of this modelling. Since the entire pipeline relies on these result, it must be evaluated to ensure that binning result is not dominated by TNF values alone (this again highlights that authors must explain how they combined the Random forest and TNF vectors).

Benchmarks are performed on very old and not commonly used tools. Please evaluate against algorithmic tools like MetaWatt (https://www.frontiersin.org/articles/10.3389/fmicb.2012.00410/full), MaxBin 2 (https://academic.oup.com/bioinformatics/article/32/4/605/1744462), machine learning tools like VAMB/semi-bin and graph based tools like MetaCOAG (https://link.springer.com/chapter/10.1007/978-3-031-04749-7_5).

Cat Fecal Microbiome Dataset - this dataset has short-read sequencing. But the authors present MetaBAT-LR. I was under the impression that tool was developed to bin metagenomic long reads. But it seems it is a contigs binning tool and does not handle long reads at all.

Additional comments

The name of the tool is misleading. This does not use Long Reads as input. Hence cannot be pitched as a long reads tool.

Most popular contig binning tools are not included in the evaluation.

·

Basic reporting

1. The section “Self-supervised machine learning” is key but difficult to follow. Rather than describe the many numerical representations only inline within the text, it would help readers to tabulate these entities. Additionally, I would have greatly appreciated a toy matrix representation.

2. As above, the section “Bin merging and scaffold recruitment” also is key and would benefit from the addition of succinct mathematical notation and slightly more explicit language. At present, it is necessary to refer to the codebase to understand with certainty what is being calculated. An example of explicitness would be (at line 154) “The percentage of <scaffold> pairs … “

3. Figure 1: Although a workflow diagram is a frequent feature of bioinformatics articles, I feel that figure 1 is currently somewhat trivial. As a visual aid to express the major contribution of this work, a second panel could be added that distils in more detail the core operations of the algorithm currently represented between “Random Forest Model” and “Improved Metagenome Bins”. This would also help to alleviate the descriptive shortcomings of the present Materials and Methods.

4. For each of the graphs mentioned in the manuscript, both nodes and edges should be clearly defined by the nature/composition of the entities used.

5. Line 158: Please clarify the set of scaffolds used to construct the graph, as it is left to the reader to assume “all scaffolds”.

6. Line 203: Please clarify whether these are percent increases or absolute values of completeness. That is, is it a 52% increase for Lactobacillus or has it now reached 52% completeness? If it latter, then readers will need the starting completeness value as well.

Experimental design

no comment.

Validity of the findings

Line 302: Regarding the failure to improve completeness for some bins within the real datasets, one contributing reason not mentioned might be the impact of excluding a significant portion of the metagenomic assembly when imposing a minimum scaffold length of 1500 bp. This would particularly exacerbate the binning of more complex communities, where total diversity and microheterogeneity combine to have a significant deleterious effect on N50.

Reviewer 3 ·

Basic reporting

No comment

Experimental design

Despite the name, metaBAT-LR is not a standalone binner. Instead, it is a binning refiner that uses Hi-C linkage information along with other contig features such as coverages, but otherwise relies on the quality of the initial binning (that is done by metaBAT). Therefore, metaBAR-LR should be compared not only with standalone binners, but also binning refiners, especially those that may use read-conectivity information, such as METAMVGL (Zhang et al, 2021) and BinSPreader (Tolstoganov et al, 2022).

Since it is a binning refiner, then the influence of the initial binning quality on the final binning should be assessed as well. I would suggest to use at least VAMB and MetaWRAP. For example, BinSPreader paper states that refining of VAMB binning of Zymo using paired-end reads and / or hi-c data results in almost 100% F1 score (calculated out of completeness and purity) of the binning, while refining of metaBAT bins was not so successful due to some contamination of the bins. I think this could be done relatively quickly as initial binnings (as well as gold standard bin assignment) could be taken from e.g. BinSPreader supplementary.

Also, why bin3c and proxymeta results on Zymo were not shown? I think all tools should be benchmarked uniformly similar.

Validity of the findings

Standard de-facto tool for MAG quality evaluation when the final result is known is AMBER (Meyer et al, 2018).

The paper would strongly benefit if AMBER results will be shown as otherwise the results shown are a bit patchy, for example, Figure S2 shows aligned lengths, but no genome sizes were provided. Figure 2 shows completeness and contamination separately, but it would be great to see them combines in F1 score. All these (as well as per-bin statistics, graphs and MIMAG completeness criteria) could be automatically obtained from AMBER results.

---

## Round 0.2 · Major Revisions

Please evaluate and compare MetaBAT-LR with other binners such as HiCBin:(https://genomebiology.biomedcentral.com/articles/10.1186/s13059-022-02626-w), hicSPAdes-binner (https://cab.spbu.ru/software/hicspades/) etc.

Page 11, lines 327-331: Add the appropriate references to support the statement.
Page 11, line 341: (as seen in the CAMI2 challenge) - What is CAMI2 challenge? Provide appropriate reference.

·

Basic reporting

no comment

Experimental design

no comment

Validity of the findings

no comment

Additional comments

I appreciate the efforts of the authors in addressing my concerns.

Reviewer 3 ·

Basic reporting

no comment

Experimental design

Thanks for adding changes to the manuscript, the exposition seems to be better, however, I am little bit confused about some points.

For example, you're saying:
>1) All the command line parameters are listed in the materials and methods section

However, for BinSPreader you're saying that it was run "in default mode". However, default mode of BinSPreader does not utilize any additional paired-end connectivity.

Therefore, please do explicitly state all command-line options for all tools that are required for the reproduction of the results in a separate supplementary text. The necessary auxiliary files (e.g. assembly graph, scaffolds, binning tables, etc.) should also be available for the end users; please deposit them on e.g. Figshare or similar platform. You can see how similar supplementary files are deposited in MetaCoAg or BinSPreader papers.

Finally. I clearly understand the direct connection of authors to metabat tool. However, the following claim on lines 382-384 does not seem to be fair: "We anticipate that metaBAT-LR has the potential to improve the binning experiments of all metagenome binners since it is not dependent on the output of any specific binning tool (we demonstrated this with MetaBAT 2 as an example)". The documentation of the tool explicitly says that metabat2-produced bins should be utilized as input. In order to show the independence the authors should include at least one other binner to the study.

Validity of the findings

Thanks for running AMBER and providing some bits of its results. It would be helpful if AMBER resulting tables were available as-is as supplementary files and not just some chosen screenshots – they do contain much more useful information than showed in the paper. In addition to this, it's worth mentioning that some tools compared does not have explicit bin merging steps (this is at least true to bin3c and BinSPreader). Therefore the direct comparison might be misleading.

Please also include the initial binning stats to Figure 5.

---

## Author Rebuttal · Round 0.2

Dear Dr. Singh,

We thank the editor and all reviewers for their valuable comments. Please see our response below.

# Editor comments (Reema Singh)

MAJOR REVISIONS

1. For the reader with non-computational background, the authors should consider including a supplementary method with line-by-line commands and exact parameters used for the analysis. It will make their work more transparent.

While it looks like there are many steps involved in the analysis, we have done two things to help readers with a non-computational background to replicate the analysis presented in this work or analyze their own datasets.

1) All the command line parameters are listed in the materials and methods section.
2) The chosen parameters, are also been implemented in the software as default parameters, so users can either replicate our analysis or carry out their own. The software tool is fairly easy to run – just a single command. We also provided an example dataset with a command example.

While the parameters should work for similar datasets as we used in the work, users also have the option to fine-tune the parameters to obtain better results on their datasets.

2. Line 186-187; move to the method section.

Thank you for this catch. It has been moved.

3. Figure 4; this data should be in a table.

For Figure 4, since we added more data points according to reviewers' suggestions, the Venn diagram is not sufficient to show all the data. We have made the following changes (new Figure 5):

1) We added a stacked bar plot to compare MetaBAT-LR to alternatives (Proximeta, Bin3C, and BinSpreader, to show the overall binning statistics.

2) We added BinSpreader results to the Venn diagram to show the overlap among the software tools.

3) We added a table to show details.

4. The authors should consider citing the appropriate references in the discussion section.

We are not sure where in the discussion the Editor wanted to add references. We did, however, added a couple of references in the introduction and methods.
* * *
# Reviewer 1 (Anonymous)

## Basic reporting

Authors are presenting a technique of metagenomics binning by incorporating the Hi-C information. Using Hi-C information is quite abundant in metagenomics assembly but not predominantly used in metagenomics binning, especially on Long reads.

However, long reads are not directly handled by the tool. Hence, the tool belongs to the domain of contigs/scaffolds binning. The tool name could be misleading with other tools such as Mega-LR/MetaBCC-LR which are long reads binners.

We haven't seen any study that uses Hi-C in metagenome assembly. There are only a few studies that use Hi-C for metagenome binning.

The LR here stands for Long Range, which includes Hi-C, long reads, and other technologies that provide long-range information for binning. We do plan to support long reads in the near future.

Presentation of the paper is concise. However, literature study is poor. Although some work has been cited, they have not been used in the evaluation of the performance.

Again, there are not many similar studies about metagenome binning with Hi-C reads. Although this reviewer was not specific about what tools to include for performance evaluation, we added the comparison to a new tool, BinSpreader, suggested by Reviewer 3 (published recently).  In the BinSpreader study, the authors mainly compared its ability to other bin refining tools based on pair-end information. We are not aware of any working alternatives beyond the tools presented in this study.

## Experimental design

In the self supervised machine learning section, authors discuss the features used for the training. Please present the features in a tabular format for clear presentation. Please justify the choice of feature scaling used.

We added a table with the list of features used in this study (Figure 2A). Feature scaling is not important for Random Forest (see discussions on Stackoverflow: https://stackoverflow.com/questions/8961586/do-i-need-to-normalize-or-scale-data-for-randomforest-r-package). That said, other machine-learning methods may require feature scaling.

For example, (1.0e+06 abs(log2(d1/d2))) does not make sense without a proper explanation. This is an assymmetric feature, order of d1 and d2 will affect the feature value. Also constant added seem to mask the log ratio, please explain the mathematics behind this (lines 142-148).

First, there was a mistake in the formula here – thank you for pointing it out. The 1.0e+6 belongs to the normalized Hi-C counts (as we use it to avoid losing precision, effectively we normalize the Hi-C counts per kb by the product of the two contig lengths). In the revised version, this is fixed. Second, because of the absolute value in our equation, it is a symmetrical feature.

I am not fully convinced by the use of term "self-supervised". As this looks like a semi-supervised approach. Authors use labels from MetaBAT2 to train another model. Self-supervised approach should be able to train on its own using feature discrimination. If the method used Hi-C connectivity between scaffolds with assembly graph connections to train their model, that would push it towards a self supervised method. In this case, no labelled information was used. I suggest authors read more on this - https://en.wikipedia.org/wiki/Self-supervised_learning. A solid bioinformatics related example is training ML models where data points are

trained using single copy marker genes (they should be in different bins) and assembly graph connections (connected contigs/scaffolds are more likely to be in the same bin). A good read on this would be https://doi.org/10.1609/aaai.v36i4.20388 (published in AAAI 2022).

The difference between "self-supervised" and "semi-supervised" lies at whether or not labeled examples are used in the learning. In "semi-supervised" learning, some of the input examples have labels while others don't. In "self-supervised" learning, however, none of the input examples have labels. In the metagenome binning problem, none of the input scaffolds initially have labels. We leverage metaBAT2 to assign labels to some of the scaffolds, and then add Hi-C to further refine those labels (merging bins) as well as add labels to other scaffolds (recruiting). A good explanation of different types of learning can be found here: https://towardsdatascience.com/supervised-semi-supervised-unsupervised-and-self -supervised-learning-7fa79aa9247c.

The reviewer cited the RepBin paper, which is also a class of "self-supervised" learning. In that study, the authors used k-means to form an initial set of labels to the scaffolds, while using single-copy marker genes to estimate the value k.

Authors then combine the predicted "connectedness like" metric from the training with TNF vectors to decide which bins needs to be merged. Does this mean, the whole pipeline merge bins to improve completeness? Please have a look at the Das Tool program (https://www.nature.com/articles/s41564-018-0171-1). This could be a better baseline for benchmarking due to its popularity.

MetaBAT-LR leverages both TNF, Hi-C, as well as other information such as sequencing depth and integrates them via a robust machine-learning framework. Based on the machine-learning predictions, MetaBAT-LR merges incomplete bins and recruits unbinned scaffolds. The self-supervised machine learning is a key

component of MetaBAT-LR and a major innovation presented in this study. If we leave out Hi-C information (in WGS-controls in Figure 3), the performance becomes much worse on the synthetic dataset. Therefore, we argue that Hi-C information is crucial, and tools solely based on TNF(such as Das tool) would not produce compelling results.

Please explain how the random forest predictions are combined with TNF. Also, please mention the probability threshold used to create the affinity graph.
This is a good point. We added more descriptions about the merging/recruiting process to the Materials and Methods, under "*Bin merging and scaffold recruitment*".

Lines 159-160 is not clear. LPA algorithm results in a NxN probability matrix. This tells us the probability of a node (n) having a label (l). How did you use this to partition the graph?
Please see the cited reference for LPA (Raghavan, U. N., Albert, R., and Kumara, S. (2007). Near linear time algorithm to detect community structures in large-scale networks. Physical review E, 76(3):036106.), which is a greedy algorithm with near linear performance. It is not based on affinity propagation, which has a computing complexity of n-square. Another community-detection algorithm we provided is Lovain, which in our case produces inferior results than LPA.

## Validity of the findings

lines 186-192 - higher test accuracy could potentially mean an overfitting scenario. It might be more useful for the reader to evaluate the accuracy of this modelling. Since the entire pipeline relies on these result, it must be evaluated to ensure that binning result is not dominated by TNF values alone (this again highlights that authors must explain how they combined the Random forest and TNF vectors). Overfitting refers to models "memorizing" training data, and therefore may not generalize well to test data. In our machine learning experiments, we divided the datasets into training, validation, and test datasets. Training is used to build the model, validation is used to fine-tune model parameters, and test data is used to independently evaluate the final model performance. Therefore, a high accuracy on the test dataset indicates a good generalization capability of the model.

And we won't repeat the discussion of the importance of Hi-C information in the above response.

Benchmarks are performed on very old and not commonly used tools. Please evaluate against algorithmic tools like MetaWatt (https://www.frontiersin.org/articles/10.3389/fmicb.2012.00410/full), MaxBin 2 (https://academic.oup.com/bioinformatics/article/32/4/605/1744462), machine learning tools like VAMB/semi-bin and graph based tools like MetaCOAG (https://link.springer.com/chapter/10.1007/978-3-031-04749-7_5).
Many of the tools listed above do not incorporate Hi-C reads in order to improve binning. The selection of tools we chose to benchmark Metabat-LR against all tools that utilize Hi-C reads to improve binning. While the tools used for benchmarking might be less cited than some of the tools listed above, we have included a newly published tool in our benchmark. BinSPreader (suggested by Reviewer 3) was a tool that was published in August 2022 and also utilizes Hi-C reads to improve binning.

Cat Fecal Microbiome Dataset - this dataset has short-read sequencing. But the authors present MetaBAT-LR. I was under the impression that tool was developed to bin metagenomic long reads. But it seems it is a contigs binning tool and does not handle long reads at all.

The main focus of the paper was to show how long-range information, in the form of Hi-C can be used to improve metagenome binning.

Hi-C is a powerful technique used in genomics to study the three-dimensional architecture of genomes. It provides valuable long-range genomic information by revealing physical interactions between chromosomal regions that can be separated by large genomic distances (e.g., megabases).

## Additional comments

The name of the tool is misleading. This does not use Long Reads as input. Hence cannot be pitched as a long reads tool.

Most popular contig binning tools are not included in the evaluation.
* * *
# Reviewer 2 (Matthew Z. DeMaere)

## Basic reporting

1. The section "Self-supervised machine learning" is key but difficult to follow. Rather than describe the many numerical representations only inline within the text, it would help readers to tabulate these entities. Additionally, I would have greatly appreciated a toy matrix representation.

Great comment, both were added to the methods section in Figure 2.

2. As above, the section "Bin merging and scaffold recruitment" also is key and would benefit from the addition of succinct mathematical notation and slightly more explicit language. At present, it is necessary to refer to the codebase to understand with certainty what is being calculated. An example of explicitness would be (at line 154) "The percentage of <scaffold> pairs … "

We added the above suggested to the methods section, under "*Bin merging and scaffold recruitment*".

3. Figure 1: Although a workflow diagram is a frequent feature of bioinformatics articles, I feel that figure 1 is currently somewhat trivial. As a visual aid to express the major contribution of this work, a second panel could be added that distils in more detail the core operations of the algorithm currently represented between "Random Forest Model" and "Improved Metagenome Bins". This would also help to alleviate the descriptive shortcomings of the present Materials and Methods.

We added the above suggested to the methods section, please see the new Figure 1.

4. For each of the graphs mentioned in the manuscript, both nodes and edges should be clearly defined by the nature/composition of the entities used. (Lines ~156)

All references to graphs are now clearly defined.

5. Line 158: Please clarify the set of scaffolds used to construct the graph, as it is left to the reader to assume "all scaffolds".

We changed "scaffolds" to "all scaffolds" to clear the ambiguity. Thanks for pointing it out.

6. Line 203: Please clarify whether these are percent increases or absolute values of completeness. That is, is it a 52% increase for Lactobacillus or has it now reached 52% completeness? If it is the latter, then readers will need the starting completeness value as well.

We have added more clarity to this section. It is the latter case. To correct for this we have stated the starting genome completeness as well as the genome completeness levels after the increase.

## Experimental design

no comment.

## Validity of the findings

Line 302: Regarding the failure to improve completeness for some bins within the real datasets, one contributing reason not mentioned might be the impact of excluding a significant portion of the metagenomic assembly when imposing a minimum scaffold length of 1500 bp. This would particularly exacerbated the binning of more complex communities, where total diversity and microheterogeneity combine to have a significant deleterious effect on N50.

Thank you very much for pointing this out. We agree with this statement. We also believe it can be an alternative explanation for the results we are seeing in the two real-world datasets. We added this possibility to the discussion.

# Reviewer 3 (Anonymous)

## Basic reporting

No comment

## Experimental design

Despite the name, metaBAT-LR is not a standalone binner. Instead, it is a binning refiner that uses Hi-C linkage information along with other contig features such as coverages, but otherwise relies on the quality of the initial binning (that is done by metaBAT). Therefore, metaBAR-LR should be compared not only with standalone binners, but also binning refiners, especially those that may use read-conectivity information, such as METAMVGL (Zhang et al, 2021) and BinSPreader (Tolstoganov et al, 2022).

METAMVGL is a great tool that is able to use a multi-view graph-based metagenomic contig binning algorithm that integrates both assembly and paired-end graphs to group fragmented contigs into clusters. We chose to not to use this tool for comparison due to the fact that it does not utilize Hi-C reads in order to improve binning results. Thank you for introducing the newly published tool, BinSPreader. We added it to the list of tools to compare as it can utilize Hi-C reads in the refining process.

Since it is a binning refiner, then the influence of the initial binning quality on the final binning should be assessed as well. I would suggest to use at least VAMB and MetaWRAP. For example, BinSPreader paper states that refining of VAMB binning of Zymo using paired-end reads and / or hi-c data results in almost 100% F1 score (calculated out of completeness and purity) of the binning, while refining of metaBAT bins was not so successful due to some contamination of the bins. I think this could be done relatively quickly as initial binnings (as well as gold standard bin assignment) could be taken from e.g. BinSPreader supplementary.

The main point of this study is to show that long-range information, such as Hi-C, can be used in a robust machine-learning framework to refine metagenome binning results. To this end, we purposefully constructed a "sensitized" dataset, i.e., a not-so-deep Zymo Mock dataset (subsampled 20%) to mimic real-world datasets where assembly/binning is not perfect. If the assembly/binning were perfect, i.e., using the gold standard, then having Hi-C will not improve binning at all.

Nevertheless, we have seen different binners perform differently on different datasets (the CAMI2 challenge). We expect MetaBAT-LR to be used to refine binning experiments from all binners, as it does not rely on a specific binning tool (we used MetaBAT as an example).

This is still a good discussion point, we added the information above to the discussion section of the paper.

Also, why bin3c and proxymeta results on Zymo were not shown? I think all tools should be benchmarked uniformly similar.

We had used the ZymoBIOMICS Synthetic data to develop the tool, so that is why it was not initially used to compare the algorithms. We now show the results of bin3C and ProxiMeta in Figure 5.

## Validity of the findings

Standard de-facto tool for MAG quality evaluation when the final result is known is AMBER (Meyer et al, 2018).

The paper would strongly benefit if AMBER results will be shown as otherwise the results shown are a bit patchy, for example, Figure S2 shows aligned lengths, but no genome sizes were provided. Figure 2 shows completeness and contamination separately, but it would be great to see them combines in F1 score. All these (as well as per-bin statistics, graphs and MIMAG completeness criteria) could be automatically obtained from AMBER results.

Thank you for this suggestion. We have added AMBER results for the ZymoBIOMIC Synthetic dataset. It was very useful in providing additional data to compare the various binning tools. The genome sizes are now provided in the methods section.  We have also added the metrics per bin information provided by AMBER in the supplemental sections S6 to S8.

---

## Round 0.3 · accepted · Accept

I have assessed the revision and confirm that the authors have addressed the reviewers' comments. The revised manuscript is ready for publication.